# Analyzing trace gas filaments in the Ex-UTLS by 4D-variational assimilation of airborne tomographic retrievals

Annika Vogel<sup>1,2</sup>, Jörn Ungermann<sup>3</sup>, and Hendrik Elbern<sup>1,2</sup>

<sup>1</sup>Rhenish Institute for Environmental Research at the University of Cologne, Germany
 <sup>2</sup>Institute for Energy and Climate Research - Troposphere (IEK-8), Forschungszentrum Jülich, Germany
 <sup>3</sup>Institute for Energy and Climate Research - Stratosphere (IEK-7), Forschungszentrum Jülich, Germany
 *Correspondence to:* H. Elbern, he@eurad.uni-koeln.de

Abstract. This case study explores the potential for chemical state analysis at extratropical upper tropospheric – lower stratospheric (Ex-UTLS) height levels with airborne limb-images, assimilated into an advanced spatio-temporal system. The investigation is motivated by the limited capability of both, nadir- and limb-viewing satellite sensors to resolve highly filamented structures, delineated by sharp trace gas gradients on small horizontal and vertical scales. The EURAD-IM (EURopean Air

- pollution Dispersion Inverse Model) is applied as assimilation system and designed to extend the flight path confined retrievals from GLORIA (Gimballed Limb Observer for Radiance Imaging of the Atmosphere) to both, larger areas and detailed vertical structures by a tomographic flight pattern. Related potential and limitations of the method are studied with the following features applied: (i) airborne limb-imaging observations of the Ex-UTLS, (ii) spatio-temporal extension by 4-dimensional variational data assimilation, (iii) correlation between ozone and potential vorticity (PV) as an indicator of airmasses and (iv)
- anisotropic and inhomogeneous horizontal background error correlations in the Ex-UTLS, spreading information towards unobserved regions along PV isopleths. This setup demonstrated substantial improvements to basic approaches in exploring new data on the spatial extend and alignment of airmasses down to small-scale filaments in the Ex-UTLS. Tomographic observations provide detailed insight for reconstructing filamentary foldings along airmass boundaries above the tropopause during this case study.

# 15 1 Introduction

Extratropical upper tropospheric – lower stratospheric (Ex-UTLS) height levels are dominated by large gradients of trace gas mixing ratios between stratosphere and troposphere. This highly variable composition introduces a major source of uncertainty in modeling the distributions of atmospheric gases. Riese et al. (2013) showed that even small discrepancies in the strength of isentropic mixing can induce large uncertainties in the atmospheric composition of this area. As atmospheric chemical

composition has a strong impact on the radiative transfer, the knowledge of trace gas distributions like ozone or water vapor is crucial for the local radiation budget (e.g., Riese et al., 2014). The radiative effect of trace gases is also influencing the local dynamics by thermal effects. This leads to a direct interaction of chemical, thermal and dynamical aspects (Gettelman et al., 2011). In addition, the processes involved cover a large range of spatial and temporal scales, which makes observations as well as modeling of the Ex-UTLS especially challenging. In terms of in situ observations, long flight tracks covering the Pacific and

Atlantic Ocean are performed during the NASA Atmospheric Tomography (*ATom*) aircraft mission aiming for global chemical climatologies (Prather et al., 2017).

Different types of satellites with nadir- and limb-scanning instruments have been launched to remotely sense distributions of atmospheric trace gases on an operational basis. However, vertical profiles of nadir-looking sensors are too coarse to sample

- high gradients at the tropopause adequately. Although limb-viewing instruments like MIPAS (Michelson Interferometer for Passive Atmospheric Sounding, e.g., Fischer et al., 2008) and CRISTA (CRyogenic Infrared Spectrometers and Telescopes for the Atmosphere, e.g., Offermann et al., 1999) provide observations in much higher vertical resolution, they cannot resolve small-scale horizontal gradients as they appear along airmass boundaries in this region. To overcome this issue, a new limb-imaging approach was developed and realized in the airborne GLORIA (Gimballed Limb Observer for Radiance Imaging of
- the Atmosphere) instrument for the first time. Its detection technique allows for tomographic observations in an unprecedented spatial resolution which makes it able to resolve resolve filamentary structures in the Ex-UTLS (e.g., Riese et al., 2014; Ungermann et al., 2011).

While novel measurement techniques are able to provide observations of the Ex-UTLS in an adequate resolution, their spatial and temporal coverage is limited (e.g., Riese et al., 2014). Thus, the determination of extension and alignment of

- different trace gas filaments requires the knowledge of complete atmospheric or chemical states in space and time. Already Danielsen (1968) observed a high correlation between potential vorticity (PV) and ozone mixing ratios along isentropes in the UTLS. This correlation between a dynamical tracer like PV and an atmospheric gas is not obvious due to different sources and sinks (Riishøjgaard and Källen, 1997). Nevertheless, PV shows in practice good performance in identification of airmasses at the UTLS (e.g., Hoskins et al., 1985; Riishøjgaard, 1998)). Furthermore, Douglass et al. (1990) and Lary et al. (1995)
- demonstrated the potential of combining these meteorological fields with tracer observations to gain an approximation of the chemical state in this region.

Using data assimilation techniques, the information of observations and atmospheric models can be combined in an optimal way (e.g., Lahoz et al., 2010). The generation of an optimized atmospheric state estimation by data assimilation techniques is normally used as input for a prognostic calculation of the numerical model (e.g., Kalnay, 2003). This is expected to increase

the forecast skill of this model by reducing the errors induced by the uncertainty of initializing the model (e.g., Rawlins et al., 2007). However, the assimilation result provides detailed information about the atmospheric state at the time of interest. Therefore, it can also be used to analyze atmospheric states and processes from a scientific point of view.

Among a wide range of assimilation techniques, the 4-dimensional variational data assimilation method (4D-var) is an advanced spatio-temporal technique which is widely used in meteorological forecast systems (e.g., Riishøjgaard, 1996; Rawlins

et al., 2007; Berre et al., 2015; Bonavita et al., 2016), and increasingly also in chemistry transport modeling (e.g., Elbern et al., 2007; Errera et al., 2008; Emili et al., 2014). Advanced data assimilation techniques are able to obtain information about unobserved regions which are dynamically connected to the locations of observations (Talagrand and Courtier, 1987). As a smoother algorithm, 4D-var propagates signals forward and backward in time to optimize the initial model state with respect to assimilated observations. Thus, it ensures temporal consistency while forcing the model result towards the observations.

The background error covariance matrix (*BECM*) of an assimilation system contains the error characteristics of the underlying numerical model. Its main task is to balance the background state with the observations including the observation-errors (e.g., Bannister, 2008; Elbern et al., 2010). Therefore, a realistic representation of model (co-)variances is crucial for a reliable assimilation result as it directly influences the spatial impact of the optimization procedure (e.g., Riishøjgaard, 1998; Weaver

- and Courtier, 2001). The spatial consistency of a numerical model induces correlations between different elements of the model state vector. Assuming similar spatial error characteristics for highly correlated values, for example within the same airmass, the related forecast errors are also correlated (Riishøjgaard, 1998). In this case the BECM has to consider also covariances between various elements of the model state. This leads to a non-diagonal matrix, the dimension of which increases quadratically with the size of the model state vector. For common numerical models for regional or global meteorological applications
- the number of elements can become as high as of the order of 10<sup>12</sup>, which renders the matrix neither calculable nor explicitly storeable (e.g., Weaver and Courtier, 2001; Elbern et al., 2010). For that reason, the BECM is often implemented as an operator containing the background error (co-)variances.

Riishøjgaard (1998) and Kalnay (2003) indicated, that ensembles may be used to calculate covariances directly via the ensemble spread. Besides the large number of calculations needed to run a representative ensemble, the resulting (co-)variances

- are highly dependent on the ensemble creation. Additionally, the limited number of realizations induces sampling errors which require additional actions like localization (e.g., Hamill et al., 2001). This creates a need for determining correlations having minimal computational efforts but maximal physical meaning. Bannister (2008) summarized various ways of formulating nonzero error correlations via correlation modeling, which are currently used. One group of methods is based on transformation of the model parameters to control variables having diagonal error characteristics (e.g., Parrish and Derber, 1992). Another set
- of methods uses spatial transformations to approximate background error correlations. In the diffusion approach introduced by Derber and Rosati (1989) and applied to atmospheric data assimilation by Weaver and Courtier (2001), error correlations are modeled by diffusion of background variances towards neighboring locations. To be able to relate the error correlations to the actual flow, the diffusion coefficient may be increased or reduced anisotropically. Elbern et al. (2010) derived a flow-dependent stretching of the horizontal correlation length for stratospheric chemical applications based on fields of PV.
- The objective of this case study is to investigate the potential and limitations for analyzing small-scale filaments in the Ex-UTLS by chemical data assimilation. Specifically, the following features are developed and evaluated: (i) airborne GLORIA limb-imaging observations providing accurate information down to filamentary structures, (ii) spatio-temporal extension by 4dimensional variational data assimilation to allow for optimization of unobserved airmasses which are dynamically connected to observations, (iii) correlation between ozone and PV as an indicator of airmasses for model background initialization and
- (iv) anisotropic and inhomogeneous horizontal background error correlations, spreading information along PV isopleths. This paper is organized as follows: The EURAD-IM assimilation system is described in Sect. 2 including a short description of 4D-var assimilation (2.1), ozone-initialization via PV-correlation (2.2), the implementation of anisotropic background error correlations (2.3) and a short overview of the GLORIA instrument (2.4). The setup of the case-study is described in Sect. 3. Section 4 presents the results of assimilating GLORIA ozone-observations focusing on the distributions of airmasses of
- different scales. Section 5 concludes this study with a discussion of the results including an outlook.

## 2 Assimilation system

This section describes the numerical spatio-tempoal assimilation system *EURAD-IM* (*EURopean Air pollution Dispersion – Inverse Model*) and the airborne instrument *GLORIA*.

- The *EURAD-IM* model system performs chemical data assimilation of the lower atmosphere with high spatial and temporal resolution. It combines four-dimensional variational data assimilation (*4D-var*) with a state-of-the-art chemistry transport model (*CTM*, e.g., Elbern et al., 2007). The *EURAD-IM* is an Eulerian multiscale CTM providing prognostic calculations of a large number of atmospheric gases and aerosols taking into account dynamical as well as chemical effects (Hass and Memmesheimer, 1995). The model domain is created by a Lambert conformal projection. The horizontal grid is structured in an Arakawa-C grid stencil and the vertical model layers are defined by terrain-following  $\sigma$ -coordinates.
- 10 Various types of atmospheric-chemical observations can be assimilated in the EURAD-IM model system. In this study, retrieved ozone-mixing ratios from the airborne remote-sensing instrument *GLORIA* were used. In addition to observational data, the chemical data assimilation system requires fields of meteorological parameters as well as information on emissions for the entire assimilation window. The meteorology for the *EURAD-IM* is calculated by the numerical weather prediction model *WRF-ARW* (*Advanced Research WRF*, e.g., Skamarock et al., 2005).

## 15 2.1 4D-variational data assimilation

25

4D-var is an advanced spatio-temporal data assimilation technique which offers the ability to optimize parameters in space and time. The optimality of its solution is analyzed by assuming Gaussian error characteristics of observations and model forecasts applied to the Bayesian theorem (Lorenc, 1988).

After some algebraic manipulation, the cost function of 4D-var can be written as follows:

20 
$$J(\boldsymbol{x}_{0}) = \frac{1}{2} \cdot \left(\boldsymbol{x}_{0} - \boldsymbol{x}_{0}^{b}\right)^{T} \mathbf{B}^{-1} \left(\boldsymbol{x}_{0} - \boldsymbol{x}_{0}^{b}\right) + \frac{1}{2} \cdot \sum_{i=0}^{n_{time}} \left[ \left( \mathcal{H} \left( \mathcal{M}_{0,i}(\boldsymbol{x}_{0}) \right) - \boldsymbol{y}_{i} \right)^{T} \mathbf{R}^{-1} \left( \mathcal{H} \left( \mathcal{M}_{0,i}(\boldsymbol{x}_{0}) \right) - \boldsymbol{y}_{i} \right) \right] ,$$
 (1)

were  $y_i$  is a vector containing all observations valid for a model timestep  $t_i$  within the assimilation window  $[t_0, t_{n_{time}}]$ . The optimization parameters are trace gas concentrations of the initial model state  $x_0$ , starting from the initial background state  $x_0^b$  for the first iteration. The nonlinear model operator  $\mathcal{M}_{0,i}$  represents the EURAD-IM chemical transport model including the whole set of featured atmospheric and chemical processes, which is described in Elbern et al. (2007) in more detail. The nonlinear transformation between model state and observation state is included in the observation operator  $\mathcal{H}$ . Both information sources contributing to the cost function are weighted by their error covariance matrices **B** and **R** for background and observations, respectively.

The gradient of the cost function with respect to the initial model state  $x_0$  reads:

$$\nabla_{\boldsymbol{x}_0} J(\boldsymbol{x}_0) = \mathbf{B}^{-1} \left( \boldsymbol{x}_0 - \boldsymbol{x}_0^b \right) + \sum_{i=0}^{n_{time}} \left[ \mathbf{M}_{0,i}^T \mathbf{H}^T \mathbf{R}^{-1} \left( \mathcal{H} \left( \mathcal{M}_{0,i}(\boldsymbol{x}_0) \right) - \boldsymbol{y}_i \right) \right] \quad ,$$
(2)

30 with  $\mathbf{M}_{0,i}$  and  $\mathbf{H}$  being the tangential linearizations of forward model and observation operator at  $\mathbf{x}_0$  and  $\mathcal{M}_{0,i}(\mathbf{x}_0)$ , respectively. A detailed description of the theoretical derivation can for example be found in Lorenc (1988) and Kalnay (2003).

The 4D-var assimilation algorithm in EURAD-IM is implemented in a preconditioned form according to Courtier (1997):

$$J(\boldsymbol{v}_{0}) = \frac{1}{2} \cdot \boldsymbol{v}_{0}^{T} \, \boldsymbol{v}_{0} + \frac{1}{2} \cdot \sum_{i=0}^{n_{time}} \left[ \left( \mathcal{H} \left( \mathcal{M}_{0,i}(\mathbf{B}^{\frac{1}{2}} \, \boldsymbol{v}_{0}) \right) - \boldsymbol{d}_{i} \right)^{T} \mathbf{R}^{-1} \left( \mathcal{H} \left( \mathcal{M}_{0,i}(\mathbf{B}^{\frac{1}{2}} \, \boldsymbol{v}_{0}) \right) - \boldsymbol{d}_{i} \right) \right]$$
(3a)

$$\nabla_{\boldsymbol{v}_0} J(\boldsymbol{v}_0) = \boldsymbol{v}_0 + \sum_{i=0}^{n_{time}} \left[ \mathbf{B}^{\frac{T}{2}} \mathbf{M}_{0,i}^T \mathbf{H}^T \mathbf{R}^{-1} \left( \mathcal{H} \left( \mathcal{M}_{0,i}(\mathbf{B}^{\frac{1}{2}} \boldsymbol{v}_0) \right) - \boldsymbol{d}_i \right) \right] \quad , \tag{3b}$$

where the optimization variable  $v_0 := \mathbf{B}^{-\frac{1}{2}} dx_0 = \mathbf{B}^{-\frac{1}{2}} (x_0 - x_0^b)$  and  $d_i := y_i - \mathcal{H}(\mathcal{M}_{0,i}(x_0^b))$  are defined to avoid the 5 calculation of the inverse background error covariance matrix  $(\mathbf{B}^{-1} \text{ or } \mathbf{B}^{-\frac{1}{2}})$ , which is computationally prohibitive for highdimensional optimization problems. For common numerical weather prediction or chemistry transport models, the number of state elements can reach the order of  $10^6$  which makes the BECM being a  $10^6 \times 10^6$  matrix (e.g., Weaver and Courtier, 2001). The limited-memory BFGS (*L-BFGS*, e.g., Liu and Nocedal, 1989) is applied to perform the iterative minimization of the

preconditioned costfunction and its gradient given by Eq. (3). The minimization is stopped after a convergence criterion is reached. Additionally, the number of iterations is limited for computational reasons.

#### 2.2 Ozone-initialization via PV-correlation

Although the general correlation between PV and ozone has been well known for decades, the concrete factor for relating ozone to PV has to be approximated for each specific situation. For this study, *MLS (Microwave Limb Sounder*, e.g., Waters et al., 2006) satellite observations have been used to determine the correlation factor between PV and ozone. MLS profiles

15 for Europe on 26 September 2012 are available around 12 UTC. Model ozone concentrations were initialized using different correlation factors at the initial time 6 UTC. Calculating forward to 12 UTC makes the model states comparable to the MLS observations. The correlation factor of

$$O_3] = 40 \frac{\text{ppbv}}{\text{pvu}} \cdot PV \quad , \tag{4}$$

was found to fit best and was used for the background initialization later. Figure 1 shows the results for initializing the ozone
 model concentration with this correlation factor. The selected model layer is referring to the observed pressure level of 170 hPa.

The high correlation between ozone and PV is only valid for stratospheric conditions and loosing validity below the dynamical tropopause. Thus, the initial ozone fields below these altitudes should be left constant. Taking the lowest height of GLORIA observations into account, a lower limit for initialization with PV was set to 6 km height. With the chosen correlation factor

of 40  $ppbv pvu^{-1}$ , the ozone fields of all model layers higher than this limit are initialized by PV fields calculated from the meteorological forecast.

## 2.3 Anisotropic background error correlations

30

4D-var should be able to account for anisotropic and inhomogeneous spatial correlations of the model state, as controlled by dynamical processes. Following Weaver and Courtier (2001), the EURAD-IM assimilation system makes use of a *diffusion approach* to account for spatial correlations of background errors. The underlying idea is to split the background error covariance

5