# Peer review of "Analyzing trace gas filaments in the Ex-UTLS by 4D-variational assimilation of airborne tomographic retrievals"

_Atmospheric Chemistry and Physics, 2017_

## Referee Comment (RC1) · Anonymous Referee #2 · 5 Jun 2017

I regret I cannot recommend publication of the paper. That is so because it does not contain in its present form results of sufficient scientific interest for publication in a journal like Atmospheric Chemistry and Physics. In addition, it is rather poorly written, making it difficult to fully understand what conclusions can be drawn from the results that have been obtained.

The paper presents a study of assimilation of airborne observations of ozone performed in the region surrounding the tropopause (the 'Ex- upper tropospheric – lower stratospheric region'). Observations have been obtained during one flight of the GLO-RIA spectrometer, and the assimilation is performed with the EURAD-IM (EURopean

[Figure]

Air pollution Dispersion – InverseModel) system. The purpose is to produce a four-dimensional description of the ozone field, the assimilation being performed through the well-known variational approach.

The variational approach requires the a priori definition of an initial background estimate for the field to be reconstructed and of an associated error covariance matrix, meant to represent the uncertainty on that background. The background is obtained from an anterior estimate of the potential vorticity (PV) field, through use of the well-known correlation between PV and ozone concentration in the stratosphere. The associated error covariance matrix is estimated through a diffusion scheme, intended at simulating the elongation of the correlation along the direction of the motion, in which both the PV and ozone concentration are conserved (at least, that is my understanding). That leads to anisotropic correlations in the background error covariance matrix.

Two questions are treated in the paper. The impact of the use of anisotropic correlations on the one hand, and the impact of the GLORIA observations on the other. But the distinction between the two is not clearly made. Concerning the former aspect, the evidence that is presented seems to essentially consist of panels (a) and (b) of Figure 5 (with the corresponding text in ll. 11-14 of p. 10, relative to observations performed along the coast of Norway). But not significant difference is visible by the eye between panels 5a and 5b. A similar remark actually applies to a structure located east of Iceland (discussed ll. 4-10 of p. 10), about which nothing significant can be seen on Fig. 5. Panels 5c and 5d are also discussed, but it is not clear either what conclusions can be drawn from them. I just mention that the scales are different in the two panels (this is not said in the paper, which can cause some confusion in the reader's mind). The authors mention a bias in panel 5c (ll. 29-30, p. 9). Where did that bias come from ? Was is corrected, and how ? Without some explanation on those points, it is not possible to clearly understand what the authors have done, and what conclusions must be drawn from what they have done. In any case, it seems to me that no conclusion can be drawn about the use of anisotropic, rather than isotropic correlations.

Concerning the impact of the GLORIA observations on the assimilation, a clear impact is visible on Fig. 6, where the assimilation creates one filament of descending stratospheric air about altitude 13 km and latitudes 62-63N (the authors mention two filaments in the text, but only one is clearly visible). That filament does coincide with a string of observations, which shows that the assimilation draws the fields towards the observations. That is certainly what one can expect from assimilation, but says nothing as to the quality of the assimilation.

The authors actually make a number of statements that are not justified about the quality of the results they have obtained. They write for instance (ll. 21-22, p. 12) 'The combination of PV-dependent initialization and background error correlation has demonstrated the ability to optimize the chemical state of an airmass even if it is only partly observed'. Impact of combination of PV-dependent initialization and background error correlation has been demonstrated, but no 'optimization' has been. Actually, the only real measure of the quality of an assimilation is obtained by comparison against independent observations (for instance, by showing that the use of anisotropic correlations leads to better fit to independent observations than the use of isotropic ones).

The only real conclusion that I can see from the paper is that assimilating the GLORIA observations has an impact in the right direction. Nothing has been shown about the relative merits of anisotropic vs. isotropic correlations. That is not sufficient for publication in an international journal. The article must be significant enlarged before it can be accepted. That could be done for instance, as suggested by the authors at the end of Section 5, through the use of additional observations, some of which could be assimilated while others would be retained for validation.

---

## Referee Comment (RC2) · Anonymous Referee #1 · 3 Jul 2017

The authors discuss the use of GLORIA ozone measurements to study the extratropical UTLS (ex-UTLS), in particular fine-scale features such as filaments. The authors make a good case for the need to identify these fine-scale features and the difficulties of doing so from satellite measurements. The authors thus provide a good motivation for the use of data assimilation to extract this information. The authors use the 4D-Var technique.

However, the results of the paper do not warrant publication in ACP. In my view, the authors demonstrate that there is an impact from using the anisotropic approach in-

stead of the isotropic approach (Fig. 5), but fail to demonstrate whether this impact is beneficial (let alone quantify this impact) by comparison against independent data, the ultimate test of the benefit of data assimilation. This is a significant shortcoming of the paper.

I would recommend the authors extend the work in this paper, following suggestions by this referee and the other referee. This should make the paper suitable for publication in ACP.

The authors need also to address the specific points below (not exhaustive).

Specific comments:

P. 9, L. 29: Where does this negative bias come from? What do you do to address it? The authors should provide this information.

Figure 5: The differences between the isotropic and anisotropic case are small. This suggests an impact from the anisotropic case. However, this does indicate whether this impact is beneficial or not. For this, one must compare the results from the assimilation with independent data (the other reviewer made this comment too). As I see it, the authors do not do such a comparison.

P. 12, L. 8: It would be helpful if the authors identified the location of these filaments in Fig. 6b.